# miRNAs and isomiRs: Serum-Based Biomarkers for the Development of Intellectual Disability and Autism Spectrum Disorder in Tuberous Sclerosis Complex

**DOI:** 10.3390/biomedicines10081838

**Published:** 2022-07-29

**Authors:** Mirte Scheper, Alessia Romagnolo, Zein Mersini Besharat, Anand M. Iyer, Romina Moavero, Christoph Hertzberg, Bernhard Weschke, Kate Riney, Martha Feucht, Theresa Scholl, Borivoj Petrak, Alice Maulisova, Rima Nabbout, Anna C. Jansen, Floor E. Jansen, Lieven Lagae, Malgorzata Urbanska, Elisabetta Ferretti, Aleksandra Tempes, Magdalena Blazejczyk, Jacek Jaworski, David J. Kwiatkowski, Sergiusz Jozwiak, Katarzyna Kotulska, Krzysztof Sadowski, Julita Borkowska, Paolo Curatolo, James D. Mills, Eleonora Aronica

**Affiliations:** 1Department of (Neuro)Pathology Amsterdam Neuroscience, Amsterdam UMC Location University of Amsterdam, Meibergdreef 9, 1105 AZ Amsterdam, The Netherlands; m.scheper@amsterdamumc.nl (M.S.); a.romagnolo@amsterdamumc.nl (A.R.); a.iyer@erasmusmc.nl (A.M.I.); 2Department of Experimental Medicine, Sapienza University of Rome, 00161 Rome, Italy; zeinmersini.besharat@uniroma1.it (Z.M.B.); elisabetta.ferretti@uniroma1.it (E.F.); 3Internal Medicine, Erasmus MC, 3015 GD Rotterdam, The Netherlands; 4Child Neurology and Psychiatry Unit, Systems Medicine Department, Tor Vergata University, 00133 Rome, Italy; romina.moavero@uniroma2.it (R.M.); curatolo@uniroma2.it (P.C.); 5Child Neurology Unit, Neuroscience Department, Bambino Gesù Children’s Hospital, IRCCS, 00165 Rome, Italy; 6Diagnose-und Behandlungszentrum für Kinder, Vivantes-Klinikum Neukölln, 12351 Berlin, Germany; christoph.hertzberg@vivantes.de; 7Department of Neuropediatrics, Charité University Medicine Berlin, 13353 Berlin, Germany; bernhard.weschke@charite.de; 8Faculty of Medicine, The University of Queensland, Herston, QLD 4029, Australia; kate.riney@health.qld.gov.au; 9Neurosciences Unit, Queensland Children’s Hospital, South Brisbane, QLD 4101, Australia; 10Department of Pediatrics and Adolescent Medicine, Medical University of Vienna, “Member of ERN EpiCARE”, 1090 Vienna, Austria; martha.feucht@meduniwien.ac.at (M.F.); theresa.scholl@meduniwien.ac.at (T.S.); 11Motol University Hospital, Charles University, 15000 Prague, Czech Republic; borivoj.petrak@post.cz (B.P.); alice.maulisova@cmail.cz (A.M.); 12Reference Centre for Rare Epilepsies, Department of Pediatric Neurology, Necker Enfants Malades University Hospital, APHP, Member of ERN EpiCARE, Université de Paris, 149 Rue de Sèvres, 75015 Paris, France; rimanabbout@yahoo.com; 13Department of Translational Neurosciences, University of Antwerp, 2000 Antwerp, Belgium; anna.jansen@uzbrussel.be; 14Department of Child Neurology, Brain Center University Medical Center, Member of ERN EpiCare, 3584 BA Utrecht, The Netherlands; f.e.jansen@umcutrecht.nl; 15Department of Development and Regeneration Section Pediatric Neurology, University Hospitals KU Leuven, 3000 Leuven, Belgium; lieven.lagae@uzleuven.be; 16Department of Neurology and Epileptology, The Children’s Memorial Health Institute, 04-730 Warsaw, Poland; murbanska@iimcb.gov.pl (M.U.); sergiusz.jozwiak@gmail.com (S.J.); kotulska.jozwiak@gmail.com (K.K.); sadowski.ks@gmail.com (K.S.); j.borkowska@ipczd.pl (J.B.); 17International Institute of Molecular and Cell Biology, 02-109 Warsaw, Poland; atempes@iimcb.gov.pl (A.T.); mblazejczyk@gmail.com (M.B.); jaworski@iimcb.gov.pl (J.J.); 18Department of Medicine, Brigham and Women’s Hospital, Boston, MA 02115, USA; dk@rics.bwh.harvard.edu; 19Department of Child Neurology, Medical University of Warsaw, 02-097 Warsaw, Poland; 20Department of Clinical and Experimental Epilepsy, UCL Queen Square Institute of Neurology, London WC1E 6BT, UK; 21Chalfont Centre for Epilepsy, Chalfont St Peter SL9 0RJ, UK

**Keywords:** epilepsy, tuberous sclerosis complex, biomarkers, serum, autism spectrum disorder, intellectual disability

## Abstract

Tuberous sclerosis complex (TSC) is a rare multi-system genetic disorder characterized by a high incidence of epilepsy and neuropsychiatric manifestations known as tuberous-sclerosis-associated neuropsychiatric disorders (TANDs), including autism spectrum disorder (ASD) and intellectual disability (ID). MicroRNAs (miRNAs) are small regulatory non-coding RNAs that regulate the expression of more than 60% of all protein-coding genes in humans and have been reported to be dysregulated in several diseases, including TSC. In the current study, RNA sequencing analysis was performed to define the miRNA and isoform (isomiR) expression patterns in serum. A Receiver Operating Characteristic (ROC) curve analysis was used to identify circulating molecular biomarkers, miRNAs, and isomiRs, able to discriminate the development of neuropsychiatric comorbidity, either ASD, ID, or ASD + ID, in patients with TSC. Part of our bioinformatics predictions was verified with RT-qPCR performed on RNA isolated from patients’ serum. Our results support the notion that circulating miRNAs and isomiRs have the potential to aid standard clinical testing in the early risk assessment of ASD and ID development in TSC patients.

## 1. Introduction

Tuberous sclerosis complex (TSC) is an autosomal dominant genetic disorder with an incidence of 1:5800 that affects many organs and systems throughout the body, including the central nervous system (CNS) [1]. Clinically, TSC is characterized by a broad spectrum of symptoms, including lymphangioleiomyomatosis, retinal abnormalities, intracardiac rhabdomyomas, renal angiomyolipomas, hypomelanotic macules, epilepsy, cortical brain malformations, and neuropsychiatric disorders, depending on the underlying genetic mutation [1]. At the molecular level, TSC is characterized by the underlying heterozygous mutation of either *TSC1* or *TSC2* gene encoding hamartin and tuberin, respectively, two proteins forming the TSC1-TSC2 complex. A loss-of-function mutation in either one of these genes leads to complete disruption of the aforementioned protein complex, inducing hyperactivation of the mammalian target of rapamycin (mTOR) pathway [2]. mTOR regulates cell growth and proliferation, protein synthesis, and metabolism, therefore the impact of this dysfunction on brain development is extensive. Multiple cortical tubers and/or radial migration lines, subependymal nodules (SEN), and subependymal giant cell astrocytoma (SEGA) represent the main neuropathological lesions in the brain of individuals affected by TSC [1,3]. Cortical tubers are considered the main origin of epileptic seizures and also show a strong association with neuropsychiatric disorders [4].

Tuberous-sclerosis-associated neuropsychiatric disorders (TANDs) include autism spectrum disorder (ASD) (present in 40–50% of individuals), attention deficit hyperactivity disorder (ADHD) (30–50%), and psychomotor delay/intellectual disability (ID) (seen in over 40–50% of individuals) [5,6,7,8]. Previously, an age-dependent association between epilepsy and ID in individuals with TSC was shown; more specifically, the frequency and the severity of ID increases when epilepsy occurs early in infancy [9]. Furthermore, early onset of seizures (before ages 2–5 years old) was shown to be significantly associated with the risk of developing ASD (81%) [9]. Overall, individuals with TSC and ASD account for 1–4% of all ASD cases [5,10,11]. The variable risk of developing ASD or ID is possibly affected by alterations in synaptogenesis and synaptic pruning, connectivity, and long-term potentiation in the mTOR pathway [12,13]. Thus, mTOR hyperactivation can be considered causative of both epileptogenesis and TANDs in TSC patients [14,15].

A biomarker is, by definition, objective and a quantifiable characteristic of biological processes. Biomarkers are known to be a measure of a response that can be functional and physiological, biochemical at the cellular level, or a molecular interaction [16]. Ideally, a biomarker should be specific, sensitive, reproducible, predictive, and accurate in distinguishing between two groups, while remaining non-invasive to the patient [17,18,19]. With the debilitating effect that TANDs have on individuals with TSC and their caregivers, it is of great importance to develop robust biomarkers, allowing for more precise and individualized medical care to help identify individuals at risk of developing TANDs before its clinical manifestation.

MicroRNAs (miRNAs) are small regulatory non-coding RNAs, 20–25 nucleotides long. These small molecules are involved in most, if not all biological processes, regulating the expression of more than 60% of all protein-coding genes in humans [20]. In most cases, miRNAs interact with the 3′ UTR of their target mRNAs to induce mRNA degradation and translational repression [21]. However, interaction with regions, including the 5′ UTR, coding sequence, and gene promoters, has also been reported [22]. Mature miRNAs are defined by their unique sequences; however, next-generation sequencing (NGS) has revealed several variations at the ends or within the mature miRNA sequence. These variations are known as isomiRs and are thought to be the result of alterations in the maturation process of miRNAs or due to the addition of nucleotides by nucleotide transferases [23]. Multiple studies have shown the potential role of miRNAs as biomarkers of various pathologies in the brain, including TSC and psychiatric disorders [24,25,26,27,28]. Previously, it has been shown that miR-34 is differentially expressed in autopsy samples, as well as peripheral blood in patients with ASD [29]. However, the knowledge on isomiRs is less extensive due to difficulties in the identification and detection [30,31,32,33]. Recently, miRNA and isomiR levels in different cell types and body fluids, such as serum and plasma, have received increased attention as non-invasive biomarkers or as indicators for diagnosis and prognosis of complex diseases [19,34]. Studies have reported that serum/plasma miRNAs are possibly released through passive leakage from damaged cells caused by inflammation, tissue injury, or apoptosis. Other mechanisms of release include active secretion through microvesicles [35,36,37], such as exosomes or active release without microvesicles through RNA-binding protein dependent pathways [38]. Furthermore, a recent study has reported correlated patterns between blood-and brain-expressed miRNAs, indicating a potential use of blood-based miRNA profiling for the investigation of miRNA activity in the brain [39]. Nevertheless, further investigation is required on the correlation between miRNA dysregulation in the brain and their expression in serum.

Given the broad spectrum of TANDs at a behavioral, psychiatric, intellectual, and neuropsychological level and the limitations of current clinical diagnostic methods, there is an urgent need to identify robust non-invasive biomarkers for individuals with TSC. In order to address this issue in the current study, our objective was to determine the feasibility of exploiting serum derived circulating miRNAs and isomiRs as biomarkers for ASD and ID in infants with TSC. Following the identification of the expression levels of serum-based miRNAs and isomiRs, we selected miRNAs for the early prediction of the occurrence of ASD and ID in young TSC patients. When possible, we used patients’ serum to confirm the feasibility and the accuracy of our identification method and verify the bioinformatic predictions. Furthermore, we tried to improve the accuracy of the biomarkers by combining several miRNAs together, to initiate a new direction in early risk assessment of ASD and ID in TSC patients.

## 2. Materials and Methods

### 2.1. Cohort

The current study was performed as part of the EPISTOP project, which was a multicenter long-term, prospective study evaluating clinical and molecular biomarkers of epileptogenesis in TSC (NCT02098759). The main goal of this study was to define biomarkers from serum collected from TSC patients at an early time point (V1: at enrollment until the age of 4 months). Patients were enrolled from November 2013 to August 2016 at 10 sites. Male or female infants of age ≤ 4 months with a definite diagnosis of TSC (Appendix A) [15,40,41], without previous seizures or medication, were enrolled after informed consent of their caregivers, which was obtained in accordance with the Declaration of Helsinki. The EPISTOP study was approved by local ethical committees at all study sites. Neurodevelopment was assessed with Bayley Scales of Infant Development (BSID)-III at 6, 12, 18, and 24 months of age. Intellectual disability (ID) was defined as cognitive DQ < 70 at age 2 years. ASD risk was based on the Autism Diagnostic Observation Scale 2 (ADOS-2) score (Toddler Module). For this study, we classified patients with TSC but no neuropsychiatric comorbidity diagnosis as the control group. Therefore, patients were divided into four groups: (1) TSC patients without TANDs (Control, *n* = 30); (2) TSC patients with ID (ID, *n* = 10); (3) TSC patients with signs of ASD alone (ASD, *n* = 6); and (4) TSC patients with both ASD and ID (ASD + ID, *n* = 13) [42]. (Table 1). All analyses were performed on serum collected from TSC patients at time point V1.

### 2.2. RNA-Sequencing and Library Preparation

RNA was isolated from 59 V1 serum samples using the miRNeasy Serum/Plasma kit (Qiagen, Düsseldorf, Germany). The concentration and purity of RNA were determined at 260/280 nm using a NanoDrop 2000 spectrophotometer (Thermo Fisher Scientific, Wilmington, DE, USA). All library preparations and sequencing were completed at GenomeScan (Leiden, The Netherlands). Samples were processed for small RNA-Seq using the TruSeq Small RNA-Seq preparation kit (Illumina, San Diego, CA, USA) in accordance with manufacturers’ guidelines. In brief, small RNA was isolated from purified RNA by size selection after ligation of sequencing adapters. After gel excision, the selected RNA fragments were amplified by PCR. All clustering and DNA sequencing used the Illumina cBot and the HiSeq 4000. All samples sent for small RNA-Seq were subjected to paired-end sequencing with a read length of 151 nucleotides to a depth of 12 million reads per sample.

### 2.3. Bioinformatic Analysis

Read quality was assessed using FastQC v11.8 software produced by the Babraham Institute (Babraham, Cambridgeshire, UK), and Trimmomatic v0.36 was used to filter low-quality base calls and any adapter contamination [43]. Low quality leading and trailing bases were removed from each read, and a sliding window trimming using a window of four and a phred33 score threshold of 15 was used to assess the quality of the read body. Any reads < 17 nucleotides were discarded.

miRNA and isomiR expression levels were assessed using isomiRage [44]. The reads that passed quality control were aligned to a custom reference genome that included the sequences of all canonical mature miRNAs in accordance with miRbase v29 [45] and all possible isomiR variants. The isomiR variants were generated by including all the possible combinations of one, two, or three bases, extending the 5′- or the 3′-end of known miRNA sequences, plus the sequences obtained by trimming canonical miRNA from their 3′-end down to a length of 18 bp. Alignment to the custom genome was performed using Bowtie v1.1.2; no mismatches were allowed [46]. Only the best alignment was reported for each read. The number of reads that aligned to each canonical miRNA and isomiR was summed to give the unnormalized isomiR expression matrix. Standard small RNA-Seq alignment workflows do not distinguish between miRNA variants. To represent this, all isomiRs derived from each canonical miRNA were summed together to give the number of reads aligning to each miRNA, producing an unnormalized miRNA count matrix.

To gain an understanding of the underlying structure of the data and to search for presence of any potential confounding factors or batch effects, the data was assessed using a Principal Component Analysis (PCA), as previously described [47,48]. Next, normalization and differential expression analyses were performed on miRNAs and isomiRs that had a read count of ≥1 in any of the samples in the comparisons using the R package DESeq2 [49]. The false discovery rate was controlled for using the Benjamini-Hochberg correction, with miRNA and isomiR expression changes with an adjusted *p*-value < 0.05 being considered differentially expressed (DE).

For each comparison, a stably expressed reference gene was selected based on a covariance < 0.3 (Appendix A). Next, the expression of each miRNA and isomiR was divided by the selected reference gene to create a ratio. This ratio was used as input for a Receiver Operating Characteristic (ROC) curve analysis, and an area under the curve (AUC) was calculated. The resulting AUC showed the ability of a miRNA- or isomiR-based ratio to distinguish between two groups. The analyses were carried out in R studio, using the package pROC [50]. Finally, penalized logistic regression was performed on microRNA profiles in each pair of groups of interest to determine the best microRNA predictors using the glmnet R package [51]. The least absolute shrinkage and selection operator (LASSO) regularization was applied to find a miRNA signature minimizing the number of features. Logistic regression with the resulting signature was used to classify subjects in the groups of interest.

To assess the robustness of each miRNA/isomiR classifier, a permutation analysis was performed. For each permutation, the samples were randomly assigned to groups and ROC analysis was performed for the classifier of interest. The AUC for each permutation was recorded. This was repeated 30,000 times to produce a test statistic distribution. Finally, by observing where the original AUC fell within this distribution, the *p*-value could be calculated.

In this analysis, the results discriminated between (i) control patients v. patients with ID (controls v. ID), (ii) control patients v. patients with ASD (controls v. ASD), (iii) between control patients v. patients with ASD and ID (controls v. ASD + ID), (iv) between patients with ID v. patients with ASD and ID (ID v. ASD + ID), and (v) between patients with ASD v. patients with ASD and ID (ASD v. ASD + ID) at the V1 timepoint (≤4 months old). Every very analysis was performed for miRNAs first, followed by isomiRs. The adopted nomenclature of isomiRs used throughout this report is as follows: (i) when the isomiR sequence is followed by _miRNA (ii.e., hsa-miR-410-3p_miRNA), we refer to the canonical sequence of the isomiR; (ii) when the isomiR sequence is followed by a combination of nucleotides and _3prime (i.e., hsa-miR-409-3p_AT_3prime), we refer to the 3′ prime modification added to the canonical miRNA sequence; (iii) when the isomiR sequence is followed by a combination of nucleotides and _5prime (i.e., hsa-miR-323a-3p_G_5prime), we refer to the 5′ prime modification added to the canonical miRNA sequence; (iv); when the isomiR sequence is followed by _trimX (i.e., hsa-miR-221-3p_trim3), we refer to trimming of X nucleotides starting from the 3′ prime end of the canonical sequence of the isomiR.

### 2.4. miRNA Isolation and TaqMan Polymerase Chain Reaction (PCR) for Verification

To validate selected putative biomarkers, RT-qPCR was performed. Total RNA, including the miRNA fraction, was isolated from blood serum samples using the miRNeasy Serum/Plasma Kit (Qiagen Benelux, Venlo, The Netherlands), according to manufacturer’s instructions. The concentration of the RNA was determined at 260/280 nm using the NanoDrop 1000 Spectrophotometer (Thermo Fisher Scientific, Wilmington, DE, USA). For the evaluation of miRNA expression, 100 ng of total RNA was used to generate cDNA for hsa-miR-409-5p, hsa-miR-494-5p, and hsa-miR-214-3p, with reference gene hsa-miR-26b-3p (Thermofisher Scientific, Wilmington, DE, USA) using the TaqMan MicroRNA reverse transcription kit (Applied Biosystems, Foster City, CA, USA), according to the manufacturer’s instructions. Determination of miRNA expression was evaluated by the TaqMan micro-RNA assay (Applied Biosystems, Foster City, CA, USA) and run on a Roche Lightcycler 480 thermocycler (Roche Applied Science, Basel, Switzerland) in triplicates.

Quantification of miRNA expression was performed using LinRegPCR software (2020.2.0.1, Heart Failure Research Center, AMC, Amsterdam, The Netherlands) [52], as previously described [53]. miRNA C_T_ values were normalized using the mean expression of reference gene hsa-miR-26b-3p, and relative expression was determined.

## 3. Results

### 3.1. Differential Expression of miRNAs and isomiRs

Based on our expression cut-offs, 1765 miRNAs were identified as expressed in the serum of TSC patients. A PCA analysis was performed using this miRNA expression as an input, and no clear clustering of the ASD and ID based on the transcription profile could be identified (Appendix A).

Differential expression was performed, and two differentially expressed miRNAs were detected between the control and ID groups, of which both were overexpressed (Appendix A). Similar analyses were performed for the comparison of the control and ASD groups, between control and ASD + ID groups and between ASD and ASD + ID groups, and no differentially expressed miRNAs were detected. A final differential expression analysis was performed and resulted in the detection of 26 differentially expressed miRNAs between the ID and ASD + ID groups, of which eight were underexpressed and 18 were overexpressed (Appendix A).

Next, 18741 isomiRs were identified as expressed in TSC patients and a second set of PCA analyses was performed to determine whether these expressed isomiRs could distinguish individuals with different comorbidities. Similar to the PCA performed on miRNAs, the analysis resulted in the lack of identifiable patterns between ASD and ID (Appendix A). In the differential expression analysis, a total of six differentially expressed isomiRs were identified as overexpressed in ID compared to controls (Appendix A). When comparing the expression of isomiRs between controls and ASD, four overexpressed isomiRs were detected (Appendix A). A similar analysis was performed comparing ID and ASD + ID samples, which identified a total of seven differentially expressed isomiRs, consisting of four underexpressed and three overexpressed isomiRs (Appendix A). No differentially expressed isomiRs were identified between control and ASD + ID groups or ASD and ASD + ID groups.

### 3.2. Prognostic Performance of miRNAs in ASD and ID

The diagnostic accuracy of miRNAs, evaluated by ROC analysis, was determined for ID, ASD, and ASD + ID, as well as for discriminating between ID and ASD + ID and between ASD and ASD + ID (Figure 1). The ROC analysis revealed that the AUC ranged from 0.817 to 0.923 for single miRNA signatures. To determine whether including a panel of miRNAs would improve the diagnostic performance, we looked into the miRNA signature for each comparison. miRNAs and isomiRs were selected based on the highest AUC, sensitivity, and specificity. ROC curve analysis for the miRNAs showed 100% sensitivity and 73.3% specificity for hsa-miR-409-5p in discriminating between controls and ID, corresponding to an AUC of 0.880 (Figure 1A). We used a panel of four miRNAs (hsa-miR-409-5p, hsa-miR-1301-3p, hsa-miR-145-5p, hsa-miR-412-5p)) when comparing controls and ID, and the AUC raised to 0.9167, showing an increase of 0.0367 (Figure 1B). hsa-let-7i-3p showed a 100% sensitivity and 80% specificity corresponding to an AUC of 0.894 for distinguishing between controls and ASD (Figure 1C). A minor improvement (0.006) was present when we compared controls and ASD samples using a panel of two miRNAs (hsa-miR-423-3p, hsa-miR-1301-3p), resulting in an AUC value of 0.9 (Figure 1D). For the discrimination between controls and ASD + ID, hsa-miR-214-5p showed 84.6% sensitivity and 73.30% specificity, corresponding to an AUC of 0.817 (Figure 1E). For the comparison between ID and ASD + ID, hsa-miR-494-3p showed an 84.6% sensitivity and a 90% specificity, corresponding to an AUC of 0.908 (Figure 1F). A significantly improved result was obtained comparing ID and ASD + ID, with a signature of six miRNAs (hsa-miR-154-5p, hsa-miR-214-5p, hsa-miR-376b-3p, hsa-miR-379-3p, hsa-miR-409-5p, hsa-miR-494-3p), showing an AUC of 0.992, leading to an increase in AUC of 0.084 (Figure 1G). For the discrimination between ASD and ASD + ID, hsa-miR-103a-3p showed a 100% sensitivity and 83.3% specificity, corresponding to an AUC of 0.923 (Figure 1H). The top 10 single miRNA classifiers for each comparison are listed in Appendix A. To test the robustness of the top miRNA classifiers, a permutation analysis (*n* = 30 000) was performed. The permutation analysis demonstrated that each classifier performed better than would be expected by chance alone (*p* < 0.05, Appendix A).

### 3.3. Verification of miRNA biomarkers in ASD and ID

We further verified the predictive properties of miRNA biomarkers in TSC patients with ASD and ID using TaqMan PCR. A sub-cohort consisting of eight controls, nine ID samples, and seven ASD + ID samples was used for verification. In this verification process, we looked at three comparisons and found a sensitivity of 83.3% and a specificity of 75.0% for the discrimination between controls and ASD + ID, with an AUC of 0.792 (Figure 2A). For the comparison between ID and ASD + ID, we found an 85.7% sensitivity and 77.8% specificity, corresponding to an AUC of 0.825 (Figure 2B).

### 3.4. Prognostic Performance of isomiRs in ASD and ID

We next evaluated the performance of isomiRs to distinguish between controls, ID, and ASD + ID, as well as for discriminating between ID and ASD + ID and between ASD, and ASD + ID. A ROC analysis showed a 90% sensitivity and 80% specificity for hsa-miR-409-3p_AT_3prime in discriminating between controls and ID, corresponding to an AUC of 0.89 (Figure 3A). hsa-miR-28-3p_G_3prime showed a 100% sensitivity and 80% specificity in discriminating between controls and ASD, corresponding to an AUC of 0.922 (Figure 3B). For the discrimination between controls and ASD + ID, hsa-miR-199b-5p_miRNA showed 69.2% sensitivity and 80% specificity, corresponding to an AUC of 0.787 (Figure 3C). For the comparison between ID and ASD + ID, hsa-miR-410-3p_miRNA showed an 84.6% sensitivity and a 90% specificity, corresponding to an AUC of 0.931 (Figure 3D). Lastly, for the discrimination between ASD and ASD + ID, hsa-miR-221-3p_trim3 showed an 84.6% sensitivity and 100% specificity, corresponding to an AUC of 0.974 (Figure 3E). The top 10 single isomiR classifiers for each comparison are listed in Appendix A. To test the robustness of the top isomiR classifiers, a permutation analysis (*n* = 30,000) was performed. The permutation analysis demonstrated that each classifier performed better than would be expected by chance alone (*p* < 0.05, Appendix A).

## 4. Discussion

Approximately 80% of TSC patients experience epilepsy, and it is known as the most common neurologic symptom with seizure onset in the first two years of life. Early onset seizures present as focal seizures that may evolve into infantile spasm, which is the major cause of comorbidity and mortality in TSC [54,55,56,57]. Alongside the physical manifestations of TSC, the neuropsychiatric comorbidities represent an important feature to be further investigated. TSC-associated neuropsychiatric disorders (TANDs) can manifest with different intensities, from mild to severe, and at different levels, predominantly including behavioral difficulties (depressed mood (19–43%), anxiety (41–56%), self-injury (17–69%), aggression (37–66%), temper tantrums (47–70%), overactivity/hyperactivity (22–73%), impulsivity (36–62%), sleep difficulties (15–74%)), psychiatric disorders (autism spectrum disorder (ASD; 40–50%), attention deficit hyperactivity disorder (ADHD; 30–40%), anxiety and depressive disorder (27–56%)), and intellectual impairment (50%) [4,58,59,60,61,62,63,64]. Here, we identified a number of biomarkers, miRNAs, and isomiRs able to discriminate patients with variable neuropsychiatric comorbidities, either ASD, ID, or ASD + ID, from normally developing patients with TSC. Our most promising biomarkers were verified with RT-qPCR performed on RNA isolated from patients’ serum. This work provides a proof of concept of the relevance of small RNASeq and ROC analysis in identifying high accuracy molecular biomarkers that could be utilized for the early risk assessment of ASD and ID in TSC with the ultimate aim to improve counseling and guidance and therewith improve the quality of life of the patients and their families.

Our results demonstrated the relevant role of miRNAs (hsa-let-7i-3p and hsa-miR-409-5p) in the prediction of the early development of ASD or ID alone in TSC patients (AUC 0.894 and 0.880, respectively). Several studies have investigated the role of hsa-let-7i-3p in multiple diseases outside the brain [65,66,67,68]. On the other hand, the association between hsa-miR-409-5p and neuropsychiatric diseases, such as depression, has been reported in multiple research studies in rodents [69]. Furthermore, bioinformatic analyses revealed that target genes and pathways of hsa-miR-409-5p are involved in neurotransmitter signaling and neuroplasticity functions, further supporting its link with behavioral skills impairment and depression [69,70]. Despite presenting the lowest accuracy (AUC 0.817) overall, hsa-miR-214-5p allowed a good separation between ASD + ID and controls. However, its role in neurodevelopmental and neuropsychiatric diseases has not yet been described. Finally, our data suggested hsa-miR-103a-3p and hsa-miR-494-3p are most promising for separating TSC patients, presenting ASD + ID from patients and presenting only one of the neuropsychiatric comorbidities at the time: ASD and ID groups, respectively (AUC 0.923 and 0.908). A neuroprotective role of hsa-miR-103a-3p inhibition in Parkinson’s disease has been demonstrated previously [71]. For hsa-miR-494-3p, alterations in expression have been linked to neurodegenerative disease, including Alzheimer’s disease and Parkinson’s disease, and malignant brain tumors, such as glioma, glioblastoma, and medulloblastoma [72,73,74,75,76]. The potential role of hsa-miR-103a-3p and hsa-miR-494-3p in neurodevelopmental pathologies, such as TSC, has not yet been described, despite hsa-miR-494-3 having been described as a key regulator of endometrial receptivity through the PI3K/AKT/mTOR pathway [77].

In addition, we investigated the potential use of a panel of miRNAs to improve the diagnostic performance. Our data demonstrated an improved accuracy using multiple miRNAs signature for three out of five comparisons. These data suggest that a relevant improvement in diagnostic accuracy, specificity, and sensitivity is possible by screening for a signature of multiple miRNAs. Clinicians may consider investigating and making the first risk assessment using a single miRNA signature to exclude other pathologies; if the results are unclear, screening with a panel of multiple miRNAs provide a more accurate risk profile for a clinical diagnosis of ID and ASD in patients.

The quality of our predictions was verified in a sub-cohort of eight controls, nine ID samples, and seven ASD + ID samples. Despite the limited availability of samples, the potential use of hsa-miR-214-5p and hsa-miR-494-3p was confirmed using RT-qPCR. The verification of our bioinformatic investigation of miRNAs as potential biomarkers in TSC patients with ASD and ID suggests that the method of analysis can be reliable and that the bioinformatic framework used here shows a potential for translation of a simple prognostic biomarker-based tool to the clinic.

Similar to miRNAs, the potential role of isomiRs in predicting the development of TANDs in the early life of TSC patients was confirmed. Despite the overall better diagnostic accuracy of isomiRs compared to miRNAs, verification in patients’ samples was not possible due to technical limitations. Furthermore, no literature was found on miR-221-3p_trim3, miR-28-3p_G_3prime, or miR-409-3p_AT_3prime, given that isomiRs are unknown territory and little exploration has been carried out so far. However, since miR-199b-5p_miRNA and miR-410-3p_miRNA refer to the canonical sequence of the isomiR, more literature could be found; only miR-410-3p_miRNA has been investigated in brain pathologies such as hypoxic-ischemic brain damage and glioma as a biomarker of poor prognosis [78,79].

While the function of these miRNAs needs further investigation, the results presented in this work support the idea that circulating miRNAs and isomiRs could be utilized to improve the early diagnosis of neuropsychiatric comorbidities in TSC patients in a non-invasive manner. Nevertheless, the clinical testing leading to the diagnosis should not be limited to biomarkers-based prediction.

## 5. Conclusions

We acknowledge limitations to the interpretation of our findings, since an independent TSC cohort would be required for validation as an essential step in translation to the clinic. However, a new cohort for independent validation is currently not available. In addition, disease-free controls were not included in our study due to ethical issues related to blood sampling from otherwise healthy infants. Overall, it is important to note that patient recruitment for this study was not easy, due to the intensive visit schedule and monitoring, contributing to a dropout rate of 10% [41]. Nevertheless, the uniqueness of this cohort strengthens the importance of the results described in this study on such a rare genetic disease. When interpreting the results, it is also important to note that normal development by the age of two does not ensure normal development throughout childhood and adolescence, which would require a long-term follow-up [80,81].

Ideally, validation will confirm that serum-based miRNAs and isomiRs can aid in the early diagnosis of TANDs. The early diagnosis of neuropsychiatric comorbidities is key to preparing parents for the challenges these pathologies bring, not only in taking care of their children but also in their mental health and family dynamics. It has been demonstrated exercise-based intervention on primary symptoms of ASD has a positive influence on the quality of life. Therefore, if clinicians could predict the development of ASD and ID in TSC patients, parents could learn how to improve their children’s behavioral and social skills and reduce the impact of these comorbidities on their quality of life [15,82,83,84].

Moreover, to further understand whether these biomarkers are specific for the prediction of development of ASD and ID in TSC patients or whether they have potential as more general biomarkers of ASD and ID, it would be interesting to assess other genetic diseases with neuropsychiatric comorbidities. There is a variety of genetic disorders associated with an increased risk of ASD development, including Fragile X syndrome [85], Down syndrome [86], Duchenne muscular dystrophy [87], and neurofibromatosis type 1 [88]. Thus, it would be interesting to include non-TSC individuals with neuropsychiatric disorders to assess whether any of the biomarkers could be utilized as more general biomarkers for ASD and ID.

In conclusion, analysis of miRNA and isomiR expression patterns from this unique cohort with TSC supports the notion that circulating miRNAs and isomiRs have the potential to aid standard clinical testing in the early risk assessment of ASD and ID in TSC patients. The evidence of individual miRNAs and isomiRs as being able to discriminate between TSC patients with ASD and ID is a novel contribution to TSC literature and encourages further investigation into biomarkers for TANDs and TSC. As discussed above, it will be important for future confirmatory studies to test the hypothesis of miRNAs and isomiRs to discriminate between TANDs in TSC patients generated by these exploratory analyses. If confirmed in future studies, these miRNA signatures may aid in the prediction of manifestations of ID and ASD in TSC patients, with the possibility to apply earlier interventions, thereby improving their quality of life and of their families. Nonetheless, further investigation is still required to establish the validity of our biomarkers across an independent cohort.

## Figures and Tables

**Figure 1 biomedicines-10-01838-f001:**
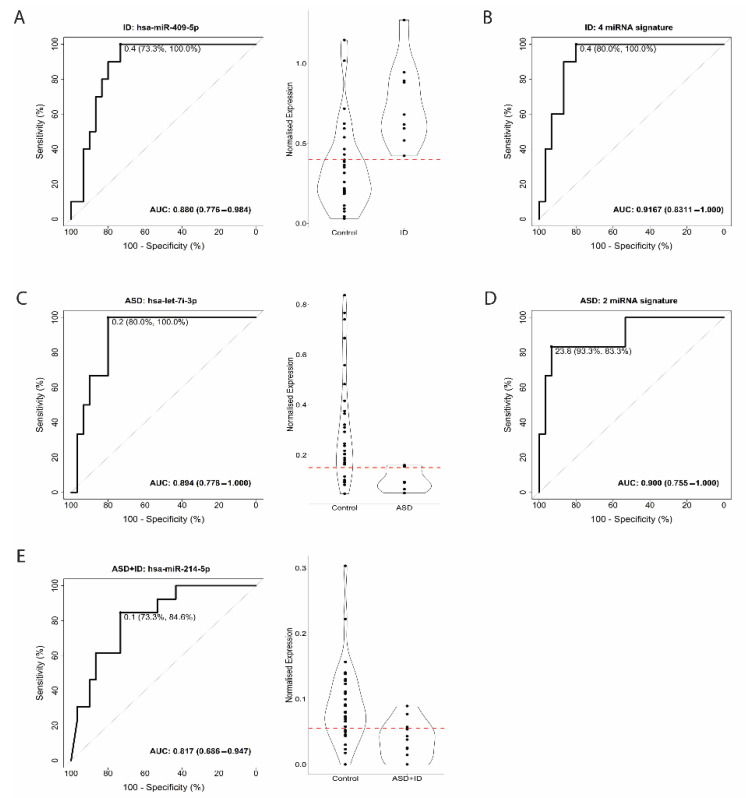
Receiver operating characteristic (ROC) curve analysis and violin plots for Intellectual Disability (ID), and Autism Spectrum Disorder (ASD), ASD + ID, as well as for discriminating between ID and ASD + ID and between ASD and ASD + ID. (**A**). ROC curve analysis for the miRNAs showed 100% sensitivity and 73.3% specificity for hsa-miR-409-5p in discriminating between controls (TSC without TANDs) and ID, corresponding to an AUC of 0.880. (**B**). Panel of four miRNAs signature discriminating between controls (TSC without TANDs) and ID. (**C**). For the discrimination between controls (TSC without TANDs) and ASD, hsa-let-7i-3p showed a 100% sensitivity and 80% specificity corresponding to an AUC of 0.894. (**D**). Panel of two miRNAs signature discriminating between controls (TSC without TANDs) and ASD. (**E**). For the discrimination between controls (TSC without TANDs) and ASD + ID, hsa-miR-214-5p showed 84.6% sensitivity and 73.30% specificity, corresponding to an AUC of 0.817. (**F**). For the comparison between ID and ASD + ID, hsa-miR-494-3p showed an 84.6% sensitivity and a 90% specificity, corresponding to an AUC of 0.908. (**G**). Panel of six miRNAs signature discriminating between ID and ASD + ID. (**H**). For the discrimination between ASD and ASD + ID, hsa-miR-103a-3p showed a 100% sensitivity and 83.3% specificity, corresponding to an AUC of 0.923. X-axis ROC curve: 100-Specificity in percentage (%); Y-axis ROC curve: Sensitivity in percentage (%); X-axis violin plot: Groups; Y-axis violin plot: Normalized expression. The red dotted line in each violin plot indicates the threshold that optimizes group separation.

**Figure 2 biomedicines-10-01838-f002:**
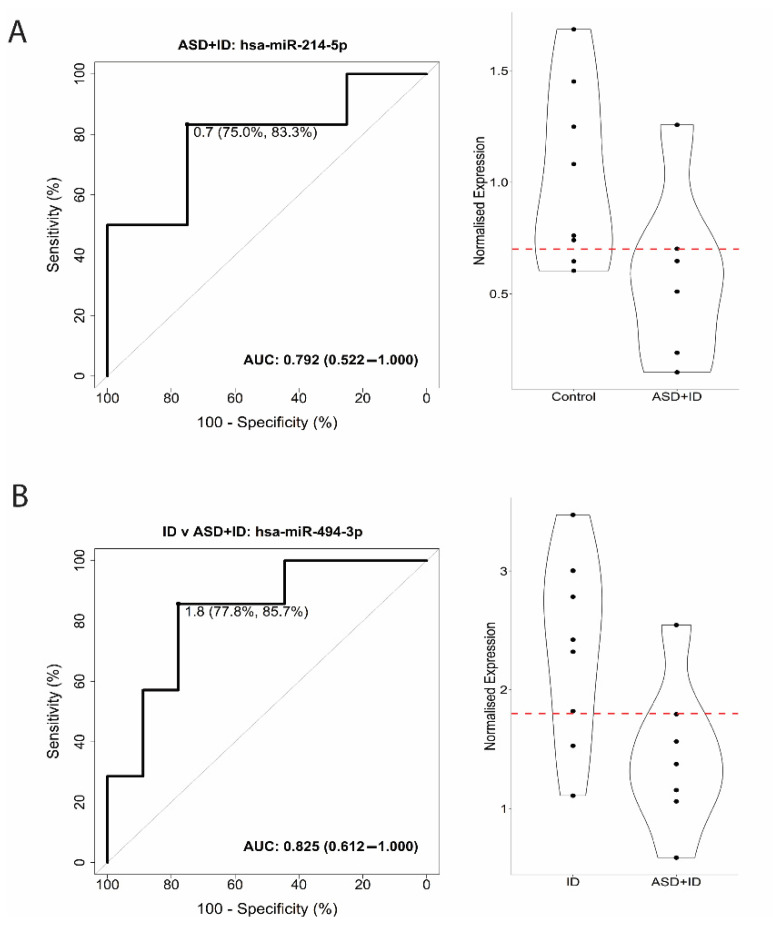
Receiver operating characteristic (ROC) curve analysis and violin plots for PCR validation of Autism Spectrum Disorder + Intellectual Disability (ASD + ID), as well as for discriminating between ID and ASD + ID. (**A**). ROC curve analysis for the miRNAs showed 83.3% sensitivity and 75.0% specificity for hsa-miR-214-5p in discriminating between controls (TSC without TANDs) and ASD + ID, corresponding to an AUC of 0.792. (**B**). For the discrimination between ID and ASD + ID, hsa-miR-494-3p showed an 85.7% sensitivity and 77.8% specificity, corresponding to an AUC of 0.825. X-axis ROC curve: 100-Specificity in percentage (%); Y-axis ROC curve: Sensitivity in percentage (%); X-axis violin plot: Groups; Y-axis violin plot: Normalized expression. The red dotted line in each violin plot indicates the threshold that optimizes group separation.

**Figure 3 biomedicines-10-01838-f003:**
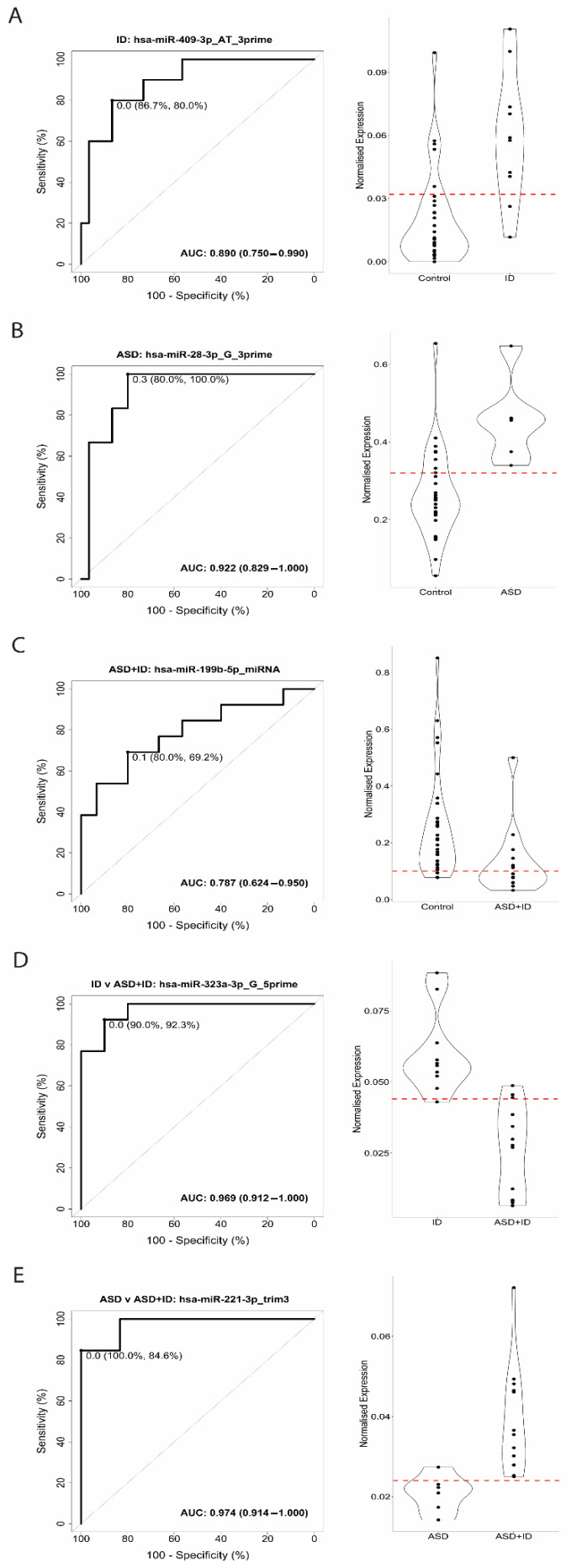
Receiver operating characteristic (ROC) curve analysis and violin plots for Intellectual Disability (ID), and Autism Spectrum Disorder (ASD), ASD + ID, as well as for discriminating between ID and ASD + ID and between ASD and ASD + ID. (**A**). ROC curve analysis for the isomiRs showed 90% sensitivity and 80% specificity for hsa-miR-409-5p for hsa-miR-409-3p_AT_3prime in discriminating between controls (TSC without TANDs) and ID corresponding to an AUC of 0.87. (**B**). hsa-miR-28-3p_G_3prime showed a 100% sensitivity and 80% specificity in discriminating between controls (TSC without TANDs) and ASD, corresponding to an AUC of 0.922. (**C**). For the discrimination between controls and ASD + ID hsa-miR-199b-5p_miRNA showed 69.2% sensitivity and 80% specificity, corresponding to an AUC of 0.787. (**D**). For the comparison between ID and ASD + ID, hsa-miR-410-3p_miRNA showed an 84.6% sensitivity and a 90% specificity, corresponding to an AUC of 0.931. (**E**). For the discrimination between ASD and ASD + ID, hsa-miR-221-3p_trim3 showed an 84.6% sensitivity and 100% specificity, corresponding to an AUC of 0.974. X-axis ROC curve: 100-Specificity in percentage (%); Y-axis ROC curve: Sensitivity in percentage (%); X-axis violin plot: Groups; Y-axis violin plot: Normalized expression. The red dotted line in each violin plot indicates the threshold that optimizes group separation.

**Table 1 biomedicines-10-01838-t001:** Baseline of the study cohort and neurodevelopmental outcome at 24 months.

	Control (*n* = 30)	ID (*n* = 10)	ASD (*n* = 6)	ASD + ID (*n* = 13)
**Mean age V1 (days)**	47.1	20.2	26.3	54.2
**Sex**				
Female	15 (50%)	7 (70%)	1 (17%)	4 (31%)
Male	15 (50%)	3 (30%)	5 (83%)	9 (69%)
**TSC mutation**				
TSC1	7 (23%)	2 (20%)	3 (50%)	1 (8%)
TSC2	22 (74%)	8 (80%)	3 (50%)	12 (92%)
NMI	1 (3%)			
**Seizures**				
Yes	17 (57%)	10 (100%)	5 (83%)	12 (92%)
No	13 (43%)	0 (0%)	1 (17%)	1 (8%)

Abbreviations: Tuberous sclerosis complex (TSC), no mutation identified (NMI), intellectual disability (ID), autism spectrum disorder (ASD).

## Data Availability

The data presented in this study are available on request from the corresponding author.

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
