# Peer review of "miRNAs and isomiRs: Serum-Based Biomarkers for the Development of Intellectual Disability and Autism Spectrum Disorder in Tuberous Sclerosis Complex"

_biomedicines, 2022, doi:10.3390/biomedicines10081838_

Round 1

Reviewer 1 Report

In this revision, the authors have incorporated my review comment and it seems to be publishable. However, before it can be accepted,  the authors should reconsider the materials presented in the discussion and conclusion sections, which can be improved to some extent. For instance, addressing the limitations and the outline of the future work should go to the conclusion section. 

So, I suggest that the authors spend some time re-organizing the text in these two sections.

Reviewer 2 Report

In the manuscript “miRNAs and isomiRs: serum-based biomarkers for the development of Intellectual Disability and Autism Spectrum Disorder in Tuberous Sclerosis Complex”, Mirte Scheper and co-workers investigates about circulating miRNAs and isomiRs as biomarkers, highlighting their po-tential in the early risk assessment of autism spectrum disorder (ASD) and intellectual disability (ID) in Tuberous sclerosis complex patients (TSC).

In my opinion, the manuscript is original and deals with an interesting thematic in a comprehensive manner. The experimental design is rich and well-set. Therefore, the paper can be considered for publication, though after addressing minor points:

1)    The abstract structure appears unbalanced in favour of a broad introduction, without the appropriate space for the discussion of methods and the conclusion.

2)    In line 71 Authors should add one or more references.

3)    In line 79 Authors should add one or more references.

4)    The Discussion part should be improved, despite being nicely set.

Author Response

This manuscript is a resubmission of an earlier submission. The following is a list of the peer review reports and author responses from that submission.

Round 1

Reviewer 1 Report

The article is well written and nicely presented to facilitate the reading, however, there are still some minor things that need to be considered:

1 The methodology section must be carefully re-written by incorporating additional explanation and justification on the proposed method.  For instance, why PCA was used, not other dimension reduction approaches?

2 In 2.4, PCR was mentioned but within the section, there is no definition or explanation on why they are used and what is PCR. mentioning the software name LinRegPCR is not so meaningful in this regard. I guess that PCR means principal component regression. 

3 Also, what is the novelty from a methodology perspective? it seems that there is no innovation in methodology.

4 Authors should consider extending the conclusion section by giving a more detailed plan for future work.

Reviewer 2 Report

This is a potentially interesting investigation to disclose the possible blood biomarkers of intellectual disability, or autism spectrum disorder, or attention deficit hyperactivity disorder, in patients affected by tuberous sclerosis complex (TSC). Apart from the small cohort examined, I believe that the major problem of the study is the lack of appropriate control groups consisting of subjects with the mentioned neuropsychological disorders but not affected by TSC. This makes it impossible to understand if the identified biomarkers are just specifically referred to TSC-associated neuropsychological disorders or, instead, these biomarkers could have a more general meaning.

Round 2

Reviewer 2 Report

Authors confirmed my criticism, the problem of this work is that it cannot give an answer to posed questions because the appropriate controls are missing.